# CorefPrompt: Prompt-based Event Coreference Resolution by Measuring Event Type and Argument Compatibilities

**Sheng Xu, Peifeng Li** and **Qiaoming Zhu**
School of Computer Science and Technology, Soochow University, China
sxu@stu.suda.edu.cn, {pfli, qmzhu}@suda.edu.cn

## Abstract

Event coreference resolution (ECR) aims to group event mentions referring to the same real-world event into clusters. Most previous studies adopt the "encoding first, then scoring" framework, making the coreference judgment rely on event encoding. Furthermore, current methods struggle to leverage human-summarized ECR rules, e.g., coreferential events should have the same event type, to guide the model. To address these two issues, we propose a prompt-based approach, CorefPrompt, to transform ECR into a cloze-style MLM (masked language model) task. This allows for simultaneous event modeling and coreference discrimination within a single template, with a fully shared context. In addition, we introduce two auxiliary prompt tasks, event-type compatibility and argument compatibility, to explicitly demonstrate the reasoning process of ECR, which helps the model make final predictions. Experimental results show that our method CorefPrompt[1] performs well in a state-of-the-art (SOTA) benchmark.

## 1 Introduction

Within-document event coreference resolution (ECR) task aims to cluster event mentions (i.e., triggers that most clearly indicate the occurrences of events) in a document such that all event mentions in the same cluster refer to the same real-world event. Consider the following example:

{Former Pakistani dancing girl}$_{arg_1}$ commits {suicide}$_{ev_1}$ 12 years after horrific {acid}$_{arg_2}$ {attack}$_{ev_2}$ which {left}$_{ev_3}$ {her}$_{arg_3}$ looking "not human". {She}$_{arg_4}$ had undergone 39 separate surgeries to repair {damage}$_{ev_4}$. Leapt to {her}$_{arg_5}$ {death}$_{ev_5}$ from {sixth floor Rome building}$_{arg_6}$ {earlier this month}$_{arg_7}$. {Her ex-husband}$_{arg_8}$ was {charged}$_{ev_6}$ with {attempted murder}$_{arg_9}$ in {2002}$_{arg_{10}}$ but has since been {acquitted}$_{ev_7}$.

This example contains seven event mentions ($ev_1$-$ev_7$) and ten entity mentions ($arg_1$-$arg_{10}$) that serve as arguments. Among them, the death event mention $ev_1$ with the argument $arg_1$ and the death event mention $ev_5$ with the arguments $arg_5$, $arg_6$, and $arg_7$ are coreferential, as both of them describe the girl's suicide by jumping off a building; the injury event mention $ev_3$ with the arguments $arg_2$ and $arg_3$ and the injury event $ev_4$ with the argument $arg_4$ are coreferential, as both of them describe the girl's disfigurement; other event mentions are singletons. Identifying the coreference between event mentions is essential for understanding the content of the text, and is the key to aggregating information that can support many downstream NLP applications, such as event extraction (Huang and Peng, 2021), discourse analysis (Lee et al., 2020) and timeline summarization (Li et al., 2021).

Event coreference resolution is more challenging than entity coreference resolution due to the complex event structure (Yang et al., 2015), triggers and corresponding arguments are loosely distributed in the text, which needs to consider the compatibilities of multiple event dimensions, including triggers, arguments, event types, etc. Most previous work regards ECR as an event-pair classification task (Huang et al., 2019; Lu and Ng, 2021a; Xu et al., 2022), i.e., judging whether two event mentions are coreferential. Early neural methods focus on obtaining trigger representations by various encoders and then manually constructing matching features (Krause et al., 2016; Nguyen et al., 2016), while recent studies integrate event compatibility into judgments using well-designed model structures (Huang et al., 2019; Lai et al., 2021; Lu and Ng, 2021a) or directly incorporate argument information into event modeling (Zeng et al., 2020; Tran et al., 2021), alleviating noise brought by wrongly extracted or empty event slots. Other work (Kriman and Ji, 2021; Xu et al., 2022) learns ECR-aware event representations through contrastive learning

---

[1]Code is available at https://github.com/jsksxs360/prompt-event-coref-emnlp2023

or multi-level modeling. However, at least two limitations exist in the above studies.

First, most current ECR studies adopt the "encoding first, then scoring" framework, wherein they first use encoders (e.g., BERT) to encode the text and obtain event mention embeddings, and then apply a scorer to evaluate the coreference score of event pairs based on these learned embeddings. This results in the "information blocking" issue. Essentially, since the scorer solely utilizes the learned embeddings as inputs, almost the entire coreference determination relies on the event encoding. However, the event encoding is performed independently, without direct influence from coreference judgment. As a result, the encoder may not accurately capture contextual information that is crucial for ECR. Especially for coreferential event pairs with unbalanced information, the learned embeddings may significantly differ if one event has rich arguments while the other has only a vague trigger. To alleviate this problem, Kenyon-Dean et al. (2018) and Kriman and Ji (2021) constrain event modeling by attraction and repulsion losses, making coreferential events have similar representations. However, this approach still performs event modeling and coreference discrimination separately, and Xu et al. (2022) find that appropriate tensor matching can achieve similar effects. Obviously, event modeling and coreference judgment are closely associated, and even humans need to review the original text to capture detailed clues when judging coreference. To address this issue, we convert ECR into a mask language prediction task using a well-designed prompt, thus the simultaneously performed event modeling and coreference judgment can interact conveniently based on a fully shared context to improve each other.

Second, previous methods struggle to utilize human knowledge, e.g., coreferential events should have the same event type and compatible arguments, and often require designing unique model structures to guide the model to focus on the compatibility of event elements, e.g., strictly matching extracted event elements (Chen et al., 2009; Cybulska and Vossen, 2015) or softly integrating event compatibilities (Huang et al., 2019; Lai et al., 2021). These approaches rely on a large amount of training, and cannot guarantee that the model can finally capture the compatibility features or understand the association between compatibilities and coreference. In the worst cases, the automatically captured features focus on other aspects and fail to discover the association between compatibilities and coreference due to the bias of interaction features or noise in the data. In this paper, we introduce two auxiliary prompt tasks, event-type compatibility and argument compatibility, to explicitly demonstrate the inference process of coreference judgment in the template and guide the model to make final decisions based on these compatibilities. Benefiting from prompting, our method can navigate the model's focus on event type and argument compatibilities using templates in natural language and can be adjusted to different compatibility levels with the assistance of soft label words.

We summarize our contributions as follows:

- Our prompt-based method CorefPrompt transforms ECR into an MLM task to model events and judge coreference simultaneously;

- We introduce two auxiliary prompt tasks, event-type compatibility and argument compatibility, to explicitly demonstrate the reasoning process of ECR.

## 2 Related Work

**Event Coreference Resolution** Event coreference resolution is a crucial information extraction (IE) task. Except for a few studies applying clustering methods (Chen and Ji, 2009; Peng et al., 2016), most researchers regard ECR as an event-pair classification problem and focus on improving event representations. Due to the complex event structure, ECR needs to consider the compatibilities of multiple event dimensions, including triggers, arguments, event types, etc. Early work builds linguistical matching features via feature engineering (Chen et al., 2009; Liu et al., 2014; Krause et al., 2016), while recent studies incorporate element compatibility into discrimination softly by designing specific model structures (Huang et al., 2019; Lai et al., 2021; Lu and Ng, 2021a,b) or enhancing event representations (Zeng et al., 2020; Tran et al., 2021). In particular, Huang et al. (2019) first train an argument compatibility scorer and then transfer it to ECR. Tran et al. (2021) build document graph structures and enhance event embeddings by capturing interactions among triggers, entities, and context words. Other methods learn ECR-aware event representations through contrastive learning or multi-level modeling (Kriman and Ji, 2021; Xu et al., 2022). For example, Xu et al. (2022) introduce full-text encoding and an event topic model

to learn document-level and topic-level event representations. However, they model events and judge coreference separately, and cannot leverage rich human-summarized rules to guide the prediction. Our approach transforms ECR into a mask language prediction task, simultaneously performing event modeling and coreference judgment on a shared context and guiding the model by explicitly showing the inference process in the template.

**Prompt-based Methods for IE** Recently, prompt-based methods (Brown et al., 2020) that convert tasks into a form close to the pre-trained tasks have attracted considerable attention. Many recent IE studies (Lu et al., 2022) explore prompt-based methods, including named entity recognition (Cui et al., 2021), relation extraction (Chia et al., 2022; Chen et al., 2022; Yang and Song, 2022), event extraction (Li et al., 2022; Ma et al., 2022; Dai et al., 2022), and event causality identification (Shen et al., 2022). These methods either transform the task into a cloze-style MLM problem or treat it as a generative problem. In particular, Chen et al. (2022) inject knowledge into the prompt by constructing learnable virtual tokens containing label semantics. Hsu et al. (2022) construct a prompt containing rich event information and then parse the elements. However, except for Shen et al. (2022) exploring event causality recognition relying on logical reasoning, most studies directly leverage PTM (Pre-Trained Model) knowledge using schema. Therefore, constructing prompts for complex IE tasks remains an under-researched problem.

## 3 Model

Formally, given a sample $x = \big(D, (ev_i, ev_j)\big)$, where $D$ is a document and $(ev_i, ev_j)$ is an event mention pair in $D$. An ECR model first needs to judge the label $y \in Y = \{\text{Coref}, \text{Non-Coref}\}$ for every event mention pair, and then organize all the events in $D$ into clusters according to the predictions. Therefore, the key lies in the coreference classification of event mention pairs. Different from fine-tuning methods that classify based on learned event mention embeddings, the prompt-based methods we adopt reformulate the problem as a mask language modeling task.

### 3.1 Prompt-based method for ECR

The most popular prompting is to convert classification into a cloze-style MLM task using template $\mathcal{T}(\cdot)$ and verbalizer $v$, where $v : Y \to V_y$ is the mapping that converts category labels to label words. Specifically, for each sample $x$, prompt-based methods first construct an exclusive template $\mathcal{T}(x)$ containing a [MASK] token, and then send it into a PTM $\mathcal{M}$ to predict the score of each label word filling the [MASK] token, and calculate the probability of the corresponding category $p(y|x)$:

$$
\begin{aligned}
p(y|x) &= p\big([\text{MASK}] = v(y)|\mathcal{T}(x)\big) \\
&= \frac{\exp P_{\mathcal{M}}\big(v(y)|\mathcal{T}(x)\big)}{\sum_{i=1}^{k} \exp P_{\mathcal{M}}\big(v(i)|\mathcal{T}(x)\big)}
\end{aligned}
\tag{1}
$$

where $P_{\mathcal{M}}\big(t|\mathcal{T}(x)\big)$ represents the confidence score of $t$ token that fills the mask position in the template $\mathcal{T}(x)$, predicted by the MLM head of $\mathcal{M}$.

In this paper, we adopt prompting to solve the two issues mentioned before by designing specific templates for ECR. Specifically, for the input event mention pair $(ev_i, ev_j)$, we construct three corresponding templates: the prefix template $\mathcal{T}_{pre}$, the anchor template $\mathcal{T}_{anc}$, and the inference template $\mathcal{T}_{inf}$. These templates respectively add guidance, encode events (including predicting event types), and discriminate coreference (including judging type and argument compatibilities). Then, we embed these templates into the segment hosting the two event mentions, converting the sample into a prompt containing multiple [MASK] tokens. Since all event tasks are completed simultaneously in the same template, multiple steps, such as event modeling and coreference judgment, can interact conveniently based on a fully shared context, reducing the "information blocking" problem that existed in previous work. Finally, we send the entire prompt to a PTM and use the PTM's MLM head to obtain the results of all tasks by predicting mask tokens. We guide PTM encoding with the prefix template and explicitly demonstrate the reasoning process of coreference judgment with the inference template, fully incorporating human knowledge into the model predictions. The overall framework of our CorefPrompt is shown in Figure 1.

### 3.2 ECR-aware Sequence Encoding

Considering that ECR is a complex task requiring reasoning, we construct our prompt by mixing multiple templates, similar to PTR (Han et al., 2022). First, following the common practice of prompting, we add a prefix template $\mathcal{T}_{pre}$ that contains knowledge before the input text to guide the model:

$\mathcal{T}_{pre}$: In the following text, the focus is on the events expressed by [E1S] $ev_i$ [E1E] and [E2S]

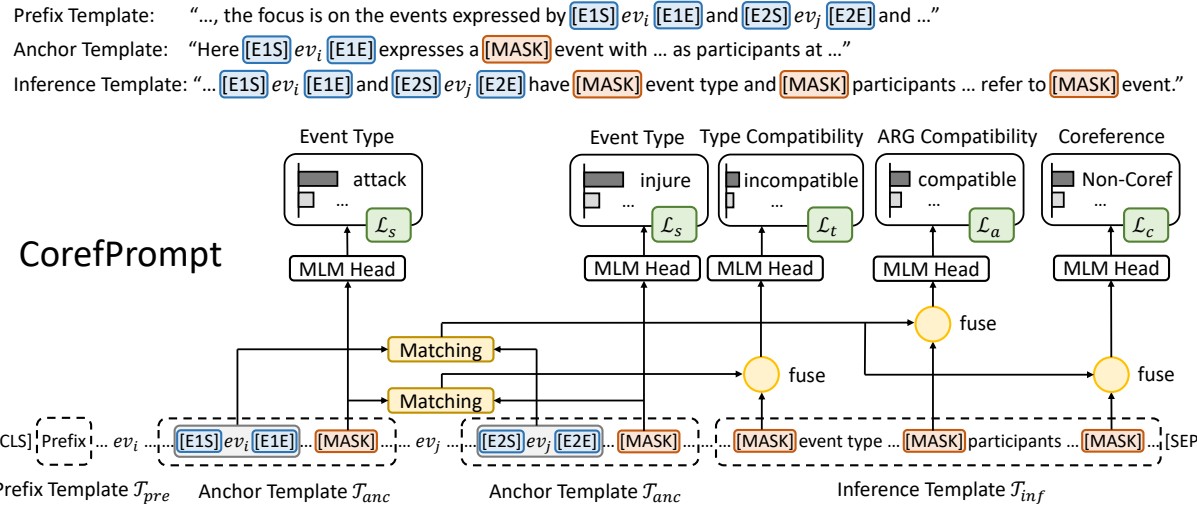

Figure 1: The overall framework of our prompt-based method CorefPrompt. We first utilize a prefix template $\mathcal{T}_{pre}$ to inform PTM what to focus on when encoding, then mark event types and arguments by inserting anchor templates $\mathcal{T}_{anc}$ around event mentions, and finally demonstrate the reasoning process of ECR using an inference template $\mathcal{T}_{inf}$ which introduces two auxiliary prompt tasks, event-type compatibility and argument compatibility.

$ev_j$ [E2E], and it needs to judge whether they refer to the same or different events.

where [E1S]/[E1E] and [E2S]/[E2E] are special event markers to represent the start/end of two event triggers $ev_i$ and $ev_j$, which will then be added to the vocabulary as learnable tokens. The prefix template first asks the model to spend additional attention on the two triggers to be processed when encoding the entire input, and then informs the model of the final objective to help the model learn embeddings relevant to the ECR task.

Previous work (Huang et al., 2019; Otmazgin et al., 2022) shows that argument compatibility and mention type are key cues to judging coreference. Therefore, we insert templates containing this information around event mentions, raising the model's attention to arguments and event types. Since these inserted templates can be seen as marking the event positions in the document, we call them anchor templates $\mathcal{T}_{anc}$. Specifically, let the argument set of event mention $ev_i$ be $A^i = P^i \cup L^i$, where $P^i = \{p_1^i, p_2^i, ...\}$, $L^i = \{l_1^i, l_2^i, ...\}$ are the participant set and location set, respectively. The corresponding anchor template (the anchor template form of $ev_j$ is same as that of $ev_i$) is:

$\mathcal{T}_{anc}$: Here [E1S] $ev_i$ [E1E] expresses a [MASK] event with $p_1^i, p_2^i, ...$ as participants at $l_1^i, l_2^i, ...$

Here, we introduce a derived prompt task to predict event types without inputting the recognized ones. Given that an event type may contain multiple tokens, it is not easy to directly find a suitable

word from the vocabulary as its label word. Thus, we create virtual label words for event types using the semantical verbalizer (Chen et al., 2022; Dai et al., 2022). Specifically, for each event type $s$, we take its tokenized sequence $\{q_1^s, q_2^s..., q_m^s\}$ as the semantic description, and then initialize the embedding of the corresponding label word $l_s$ as:

$$\boldsymbol{E}(l_s) = \frac{1}{m}\sum_{i=1}^{m} \boldsymbol{E}(q_i^s) \qquad (2)$$

where $\boldsymbol{E}(\cdot)$ represents the word embedding table, and then we expand the MLM head of PTM using these label words. During training, we calculate the cross-entropy loss $\mathcal{L}_s$ between the predicted probability distribution on label words and annotated event types as the supervision signal. In the prediction stage, we just need to keep these positions as [MASK]s. The anchor templates explicitly mark the event types and arguments and are inserted near the event mention. Therefore, benefiting from the Attention mechanism, PTM can focus more on the argument information when encoding the context to support subsequent prompt tasks.

### 3.3 Joint Reasoning with Auxiliary Tasks

After guiding the model to spend more attention on the event mention pair $(ev_i, ev_j)$ using the prefix and anchor templates, the easiest way to finish ECR is to concatenate a template $\mathcal{T}_{ECR}$, as shown follows, that converts the task into a mask language prediction problem and chooses

$Vy = \{\text{same, different}\}$ as the label word set. In this way, we can obtain the results by comparing the predicted confidence scores of these label words at the mask position.

> $\mathcal{T}_{ECR}$: In the above text, events expressed by [E1S] $ev_i$ [E1E] and [E2S] $ev_j$ [E2E] refer to [MASK] event.

However, since the complex ECR task requires the model to measure the compatibility of multiple event dimensions, it is quite difficult for the PTM to predict coreference merely by encoding the content in the prompt. Inspired by Chain of thought (CoT) prompting (Wei et al., 2022; Qiao et al., 2022), which guides the model to make the final prediction by demonstrating the reasoning process step by step, we construct an inference template $\mathcal{T}_{inf}$ to explicitly show the reasoning process of ECR by introducing two auxiliary prompt tasks:

> $\mathcal{T}_{inf}$: In conclusion, the events expressed by [E1S] $ev_i$ [E1E] and [E2S] $ev_j$ [E2E] have [MASK] event type and [MASK] participants, so they refer to [MASK] event.

As shown above, $\mathcal{T}_{inf}$ contains three prompt tasks: event-type compatibility, argument compatibility, and coreference judgment. As such, PTM can capture the associations among the predictions of these three prompt tasks to improve ECR. Benefiting from the CoT-style template form, PTM can perform deeper and more thinking than direct prediction. Considering that "compatibility" can be expressed by numerous words in the vocabulary, we again use the semantical verbalizer, i.e., Eq. (2), to create virtual label words for "compatible" and "incompatible" labels. Here, we use the manually constructed word set $V_m = V_{comp} \cup V_{incom}$ to initialize the label word embeddings: $V_{comp} = \{\text{same, related, relevant, similar, matching, matched}\}$, $V_{incom} = \{\text{different, unrelated, irrelevant, dissimilar, mismatched}\}$, where $V_{comp}$ and $V_{incom}$ correspond to the labels "compatible" and "incompatible", respectively.

### 3.4 Mask Token Embedding Updating

Previous fine-tuning work shows that tensor matching can effectively capture the semantic interactions of events (Xu et al., 2022), which is also helpful for our three prompt tasks in $\mathcal{T}_{inf}$. Therefore, we introduce similar semantic matching operations to update the mask word embeddings by injecting interactive features. In this way, the PTM can combine additional clues to predict tokens at mask positions in the inference template $\mathcal{T}_{inf}$.

Specifically, similar to traditional fine-tuning methods, we first apply the attention mechanism on top of the hidden vectors $\boldsymbol{h}_j$ of the $j$-th tokens in anchor template $\mathcal{T}_{anc}$ to obtain the embedding $\boldsymbol{e}_i$ of event mention $ev_i$ as follows:

$$\boldsymbol{e}_i = \sum_{j=p}^{q} \alpha_j \boldsymbol{h}_j \qquad (3)$$

$$\alpha_i = \frac{\exp(w_i)}{\sum_{j=p}^{q} \exp(w_j)} \qquad w_i = \boldsymbol{w}_s^\top \boldsymbol{h}_i \quad (4)$$

where $p$ and $q$ are the start and end positions of the event mention in the anchor template $\mathcal{T}_{anc}$, respectively, and $\boldsymbol{w}_s$ is the model parameter. Then, following Xu et al. (2022), we choose the element-wise product and multi-perspective cosine similarity MultiCos($\cdot$) as matching operations to capture the semantic interactions on event triggers and event types:

$$\text{M}(\boldsymbol{x}_1, \boldsymbol{x}_2) = [\boldsymbol{x}_1 \circ \boldsymbol{x}_2; \text{MultiCos}(\boldsymbol{x}_1, \boldsymbol{x}_2)] \quad (5)$$

$$\boldsymbol{m}_{Sem} = \text{M}(\boldsymbol{e}_i, \boldsymbol{e}_j) \quad \boldsymbol{m}_{Type} = \text{M}(\boldsymbol{h}_i^{Type}, \boldsymbol{h}_j^{Type})$$
$$(6)$$

where $\boldsymbol{h}_i^{Type}, \boldsymbol{h}_j^{Type}$ are the mask token embeddings used for predicting the event types of $ev_i$ and $ev_j$ in the anchor template $\mathcal{T}_{anc}$. After obtaining the trigger and event-type matching features $\boldsymbol{m}_{Sem}$ and $\boldsymbol{m}_{Type}$, we utilize them to update all the three mask token embeddings in the inference template $\mathcal{T}_{inf}$, helping the PTM finish the corresponding prompt tasks better:

$$\tilde{\boldsymbol{h}}_{\text{[MASK]}}^{Type} = [\boldsymbol{h}_{\text{[MASK]}}^{Type}; \boldsymbol{m}_{Type}]\boldsymbol{W}_t \qquad (7)$$

$$\tilde{\boldsymbol{h}}_{\text{[MASK]}}^{Arg} = [\boldsymbol{h}_{\text{[MASK]}}^{Arg}; \boldsymbol{m}_{Sem}]\boldsymbol{W}_a \qquad (8)$$

$$\tilde{\boldsymbol{h}}_{\text{[MASK]}}^{Coref} = [\boldsymbol{h}_{\text{[MASK]}}^{Coref}; \boldsymbol{m}_{Sem}]\boldsymbol{W}_c \qquad (9)$$

where $\boldsymbol{h}_{\text{[MASK]}}^{Type}, \boldsymbol{h}_{\text{[MASK]}}^{Arg}, \boldsymbol{h}_{\text{[MASK]}}^{Coref}$ are mask token embeddings corresponding to the event-type compatibility, argument compatibility, and coreference discrimination tasks, respectively, and $\boldsymbol{W}_t, \boldsymbol{W}_a, \boldsymbol{W}_c$ are parameter matrices responsible for transforming tensor dimensions to that of PTM. Finally, we feed these updated mask token embeddings into the MLM head of the PTM for prediction. Before training, we construct event-type compatibility labels according to whether $ev_i$ and $ev_j$ have the same

event type, while argument compatibility and coreference prediction tasks use the coreference labels directly. To update the parameters, we calculate the cross-entropy losses between the predictions and the labels at these mask positions. Conveniently, we denote the coreference loss as $\mathcal{L}_c$, and combine the event-type compatibility and argument compatibility losses $\mathcal{L}_t, \mathcal{L}_a$ as the compatibility loss $\mathcal{L}_m$.

### 3.5 Training

To simultaneously update parameters in the PTM (including MLM head) and the matching module, we jointly perform three tasks of event-type prediction, compatibility prediction, and coreference judgment (corresponding losses are $\mathcal{L}_s, \mathcal{L}_m, \mathcal{L}_c$, respectively), and define the overall loss function as:

$$\mathcal{L} = \sum_{i \in \{s,m,c\}} \log(1 + \mathcal{L}_i) \qquad (10)$$

In this way, the optimizer can automatically regulate the balances among these three tasks by weights $\frac{1}{1+\mathcal{L}_i}, i \in \{s, m, c\}$. In addition, inspired by the trigger-mask mechanism (Liu et al., 2020; Xu et al., 2022), we propose a trigger-mask regularization to enhance the robustness of our model. Specifically, during training, we simultaneously input another template whose triggers are all masked and ask the PTM to perform the same three tasks. This forces the PTM to mine clues from the context instead of merely memorizing the mapping from trigger pairs to coreferences. When predicting, the target positions of the event-type prediction and the two auxiliary compatibility tasks are all kept as [MASK], and we only need to output the probability distribution of the label words in the mask position corresponding to the ECR task.

## 4 Experimentation

### 4.1 Experimental Settings

Following previous work (Lu and Ng, 2021c; Xu et al., 2022), we choose KBP 2015 and KBP 2016 (Mitamura et al., 2015, 2016) as the training set (LDC2015E29, E68, E73, E94, and LDC2016E64) and use KBP 2017 (Mitamura et al., 2017) as the test set. The training set includes 817 documents annotated with 22894 event mentions distributed in 14794 clusters, and the test set consists of 167 documents annotated with 4375 event mentions distributed in 2963 clusters. Following Lu and Ng (2021c), we select the same 82 documents from the training set for parameter tuning. To reduce the

computational cost, we apply undersampling on the training set, and only about 10% of the training data are finally used (details in Appendix A). For event extraction, we directly use the triggers provided by Xu et al. (2022), and then apply the OmniEvent toolkit[2] to extract the corresponding arguments and select the argument roles related to participants and locations. After obtaining the coreference predictions of all event mention pairs, we create the final event clusters using a greedy clustering algorithm (Xu et al., 2022). Finally, we report the ECR performance using the official Reference Coreference Scorer[3], which employs four coreference metrics, including MUC (Vilain et al., 1995), $B^3$ (Bagga and Baldwin, 1998), $CEAF_e$ (Luo, 2005), BLANC (Recasens and Hovy, 2011), and the unweighted average of their F1 scores (AVG).

We choose RoBERTa with open pre-trained parameters[4] as our encoder, which has 24 layers, 1024 hiddens, and 16 heads. All new learnable tokens we add to the vocabulary have 1024-dimensional embeddings. For all samples, we truncate the context around the two event mentions to make the final input sequence length not exceed 512. For the tensor matching operations, following previous work (Xu et al., 2022), we set the matching dimension and perspective number to 64 and 128, respectively, and set the tensor factorization parameter to 4. The batch size and the number of training epochs are 4 and 10, and an Adam optimizer with a learning rate of 1e-5 is applied to update all parameters. The random seed used in all components is set to 42.

### 4.2 Experimental Results

We compare our proposed CorefPrompt with the following strong baselines under the same evaluation settings: (i) the joint model **Lu&Ng2021** (Lu and Ng, 2021b), which jointly models six related event and entity tasks, and (ii) the pairwise model **Xu2022** (Xu et al., 2022), which introduces a document-level event encoding and event topic model. In addition, we build two pairwise baselines, **BERT** and **RoBERTa**, that utilize the popular BERT/RoBERTa model as the encoder. Specifically, they first feed a segment that does not exceed the maximum length of the encoder, including two event mentions, into the BERT/RoBERTa model and then obtain the two event trigger representa-

---

[2] https://github.com/THU-KEG/OmniEvent
[3] https://github.com/conll/reference-coreference-scorers
[4] https://huggingface.co/roberta-large

| Model | MUC | B3 | CEA. | BLA. | AVG |
|---|---|---|---|---|---|
| BERT | 35.8 | 54.4 | 55.6 | 36.0 | 45.5 |
| RoBERTa | 37.9 | 55.9 | 57.3 | 38.3 | 47.3 |
| Lu&Ng2021 | 45.2 | 54.7 | 53.8 | 38.2 | 48.0 |
| Xu2022 | **46.2** | 57.4 | 59.0 | 42.0 | 51.2 |
| CorefPrompt | 45.3 | **57.5** | **59.9** | **42.3** | **51.3** |

Table 1: Performance of all baselines and our Coref-Prompt on the KBP 2017 dataset.

tions $e_i, e_j$ by an attention mechanism. Finally, the combined feature vector $[e_i; e_j; e_i \circ e_j]$ is sent to an MLP to identify coreference.

Table 1 reports the performance of the four baselines and our CorefPrompt on the KBP 2017 dataset, and the results show that our proposed model achieves comparable performance to SOTA Xu2022, without using full-text level encoding, which demonstrates the effectiveness of our method in resolving event coreference. A detailed comparison of the computational efficiency of our method and Xu2022 can be seen in Appendix B.

Benefiting from joint modeling multiple event tasks and introducing the powerful SpanBERT encoder, Lu&Ng2021 performs better than BERT and RoBERTa, with a significant improvement of more than 7.3 on the MUC metric. However, all these methods model event mentions based only on segment-level context; therefore, after Xu2022 introduces document-level event encoding based on full-text context using the Longformer encoder, the ECR performance improves considerably. Nevertheless, all of these baselines employ traditional fine-tuning frameworks that have a large gap with the pre-training tasks of PTMs, neither completely leveraging the knowledge contained in PTMs nor directly incorporating human knowledge into model predictions. In contrast, our approach designs a specific prompt to motivate the potential of PTMs in the ECR task, incorporating rich manually summarized rules to guide the model; therefore, it achieves comparable performance to Xu2022, based only on segment-level coding.

## 5 Analysis and Discussion

We first analyze the differences between prompt-based and fine-tuning methods in processing the ECR task, showing the advantages of prompting with well-designed templates, and then discuss the contribution of each component in our approach through ablation experiments.

### 5.1 Prompt Tuning or Fine-tuning?

To demonstrate the effectiveness of prompt-based methods on the ECR task, we compare the following baselines, using BERT(B) and RoBERTa(Ro) as encoders: (i) Pairwise(B) and Pairwise(Ro): fine-tuning pairwise models, which first utilize the encoder and attention mechanism to obtain event representations and then send event pair features to a scorer; (ii) Prompt(B) and Prompt(Ro): common prompt methods, which convert ECR into a cloze-style MLM task by adding a template $\mathcal{T}_{ECR}$ before the original input; (iii) CorefPrompt(B) and CorefPrompt(Ro): using the prompt specially designed for ECR to stimulate the potential of PTMs in judging coreference. The results in Table 2 show that benefiting from eliminating the gap between ECR and pre-training tasks, prompt-based methods can leverage PTM knowledge to make judgments. Therefore, Prompt(B) and Prompt(Ro) achieve 1.6 and 1.1 improvements on AVG-F1 over their fine-tuning counterparts, respectively, without performing any tensor matching. However, these methods do not consider the characteristics of the ECR task and rely only on PTMs' understanding to make predictions. In contrast, our prompt adequately incorporates the critical points of ECR, e.g., argument compatibility, to provide better guidance, thus achieving the best performance.

| Model | MUC | B3 | CEA. | BLA. | AVG |
|---|---|---|---|---|---|
| Pairwise(B) | 40.2 | 53.7 | 55.0 | 35.1 | 46.0 |
| Pairwise(Ro) | 43.8 | 55.2 | 57.0 | 38.5 | 48.6 |
| Prompt(B) | 41.4 | 54.9 | 56.2 | 37.8 | 47.6 |
| Prompt(Ro) | 45.3 | 56.1 | 57.4 | 40.1 | 49.7 |
| CorefPrompt(B) | 41.5 | 55.4 | 56.9 | 38.9 | 48.2 |
| CorefPrompt(Ro) | 45.3 | 57.5 | 59.9 | 42.3 | 51.3 |

Table 2: Comparison of fine-tuning models and prompt-based methods.

To deeply compare these methods and exclude the effect of clustering, we also report the event-pair classification F1-score in Table 3: (1) ALL: results of all event mention pairs; (2) results of event mentions with different argument states: (i) NoA: neither event has an argument; (ii) OneA: only one event contains arguments; (iii) BothA: both events contain arguments.

Table 3 shows that the prompt-based methods still outperform the corresponding fine-tuning models. In particular, the prompt-based methods allure out PTM's understanding of ECR, thus helping to judge events that contain rich argument informa-

| Model | ALL | NoA | OneA | BothA |
|---|---|---|---|---|
| Pairwise(B) | 32.6 | 34.1 | 33.2 | 32.1 |
| Pairwise(Ro) | 36.4 | 36.3 | 38.2 | 36.0 |
| Prompt(B) | 34.0 | 34.0 | 35.4 | 34.2 |
| Prompt(Ro) | 38.6 | 40.2 | 40.9 | 38.5 |
| CorefPrompt(B) | 34.6 | 35.7 | 35.7 | 34.7 |
| CorefPrompt(Ro) | 40.5 | 41.2 | 42.4 | 39.5 |

Table 3: Event-pair classification results of event mentions with different argument states.

tion, Prompt(B) and Prompt(Ro) achieve an improvement of more than 2.0 in both OneA and BothA cases. This verifies that argument compatibility is a crucial point for ECR. CorefPrompt not only guides PTM to focus on the argument information but also considers event-type compatibility, thus improving the judgment in all cases compared to common prompts. This validates the effectiveness of our prompt design in capturing event-type and argument compatibilities to improve ECR.

Table 2 shows that the common prompts also achieve acceptable performance, so we construct the following templates for comparison: (i) Connect: placing [MASK] between event mentions, and using descriptions "refer to" and "not refer to" to create virtual label words; (ii) Question: question-style prompt, which asks PTM whether two events are coreferential; (iii) Soft: wrapping event mentions with learnable virtual tokens to build the template. More details can be seen in Appendix C. The results are shown in Table 4, where HasA indicates that at least one event contains arguments:

| Model | AVG | ALL | NoA | HasA |
|---|---|---|---|---|
| Normal | 49.7 | 38.6 | 40.2 | 39.7 |
| Connect | 50.0 | 38.3 | 37.8 | 39.7 |
| Question | 50.2 | 39.7 | 40.0 | 40.4 |
| Soft | 49.7 | 38.1 | 38.7 | 37.5 |
| Ours | 51.3 | 40.5 | 41.2 | 40.9 |

Table 4: Results using different prompts.

Table 4 shows that, different from the general opinion that the template form has a significant influence (Gao et al., 2021), all common templates achieve similar performance. This may be because ECR is a complex task that relies on inference, so it is challenging to construct a high-quality template, and merely changing the template form can only bring limited impact. Soft shows that adding learnable tokens does not improve ECR, and event-pair classification performance even drops. This indi-

cates that it is difficult for PTM to learn a suitable template for the complex ECR task automatically, and the template design is more dependent on manual constraints. Our approach incorporates much human knowledge, e.g., considering both event-type and argument compatibilities, thus improving judgments in both NoA and HasA cases.

## 5.2 Ablation Study

To analyze the contribution of each component in our method, we design the following ablations: (i) -Pre: removing prefix template; (ii) -Anc: removing anchor templates, obtaining event embeddings based on the triggers in the document; (iii) -Aux: removing type compatibility and argument compatibility auxiliary tasks; (iv) -Reg: removing trigger-mask regularization; (v): -TM: removing tensor matching operations, predicting based on original [MASK] tokens. The results are shown in Table 5.

| Model | MUC | B3 | CEA. | BLA. | AVG | ALL |
|---|---|---|---|---|---|---|
| Full | 45.3 | 57.5 | 59.9 | 42.3 | 51.3 | 40.5 |
| -Pre | 44.7 | 57.0 | 59.4 | 41.3 | 50.6 | 40.0 |
| -Anc | 44.5 | 56.9 | 58.8 | 41.0 | 50.3 | 38.4 |
| -Aux | 43.9 | 56.7 | 59.1 | 40.8 | 50.1 | 38.3 |
| -Reg | 46.1 | 56.6 | 58.9 | 40.6 | 50.5 | 40.3 |
| -TM | 44.7 | 57.1 | 59.3 | 41.8 | 50.7 | 39.8 |

Table 5: Results of various model variants.

Table 5 shows that removing any part will hurt the ECR performance, especially -Anc and -Aux, causing large drops in event-pair classification F1-scores of 2.1 and 2.2, respectively. This illustrates that anchor templates and auxiliary compatibility tasks have the most significant impact on our model. In contrast, -TM brings only a tiny drop of 0.6 and 0.7 in AVG-F1 and event pair F1-score. This suggests that our well-designed prompt can implicitly capture the interactions between event mentions without deliberately using the matching module.

Since the main contribution of the anchor template is to mark the argument information, we take the variant whose anchor template only includes event type as the Base model and apply two ways to add event arguments: (i) +Arg: adding all recognized arguments, i.e., our method; (ii) +Arg(B): keeping events have balanced argument information, i.e., if one event does not contain a specific argument role, the other will also not add. The event-pair results (P / R / F1) are shown in Table 6.

The results of +Arg and +Arg(B) show that after explicitly marking the arguments, improving not

| Model | NoA | OneA | BothA |
|---|---|---|---|
| Base | 38.9 / 37.2 / 38.0 | 45.2 / 37.1 / 40.7 | 40.2 / 38.6 / 39.4 |
| +Arg | 41.5 / 40.8 / 41.2 | 47.7 / 38.2 / 42.4 | 39.3 / 39.6 / 39.5 |
| +Arg(B) | 40.0 / 42.3 / 41.1 | 44.2 / 40.6 / 42.3 | 38.3 / 40.4 / 39.3 |

Table 6: Results after adding arguments.

only the judgment of events with arguments but also those without, whose performance is greatly enhanced by 3.2 and 3.1, respectively. This may be because events with rich argument information are inherently easy to judge, especially when arguments are explicitly marked. In this way, the model can focus on distinguishing those "difficult" samples without argument information. Therefore, the less argument information the sample has, the more the performance improves (NoA > OneA > BothA). Although +Arg(B) allows the model to compare event mentions evenly, this reduces the difference between events, making the model tend to judge samples as coreferential, improving the recall while gravely hurting the precision.

To evaluate the contributions of the two auxiliary prompt tasks, we take the variant whose inference template only contains the coreference task as Base and then: (i) +TypeMatch: adding event-type compatibility task; (ii) +ArgMatch: adding argument compatibility task; (iii) +Type&ArgMatch: adding these two simultaneously, i.e., our method. Table 7 shows the event-pair classification results.

| Model | ALL |
|---|---|
| Base | 39.2 / 37.5 / 38.3 |
| +TypeMatch | 39.4 / 40.4 / 39.9 |
| +ArgMatch | 41.5 / 38.1 / 39.7 |
| +Type&ArgMatch | 42.1 / 39.0 / 40.5 |

Table 7: Results after introducing auxiliary tasks.

Table 7 shows that introducing type compatibility and argument compatibility helps the model make predictions; the F1 scores increased by 1.6 and 1.4, respectively. The +TypeMatch provides the model with additional event type information beyond the original text, significantly improving recall by 2.9. While +ArgMatch guides the PTM to focus on distinguishing argument clues, resulting in a considerable improvement in precision by 2.3. Our approach introduces these two auxiliary tasks simultaneously so that they complement each other to achieve the best performance.

## 6    Conclusion

In this paper, we design a specific prompt to transform ECR into a mask language prediction task. As such, event modeling and coreference judgment can be performed simultaneously based on a shared context. In addition, we introduce two auxiliary mask tasks, event-type compatibility and argument compatibility, to explicitly show the reasoning process of ECR, so that PTM can synthesize multi-dimensional event information to make the final predictions. Experimental results on the KBP 2017 dataset show that our method performs comparably to previous SOTA, without introducing full-text level encoding. In future work, we will continue to study how to incorporate more human knowledge into the prompt-based approach for ECR.

## Limitations

Despite the simplicity and effectiveness of our approach, it still suffers from two obvious shortcomings. First, since our model adopts a pipeline framework, we need to pre-identify the triggers in the document before constructing the prompt, which inevitably leads to error propagation. Therefore, how to design a prompt that can jointly identify triggers and judge event coreference is still an unsolved problem. Second, our experiments construct samples for every event mention pair in the document, resulting in a large training set. Although we greatly reduce the training data size by under-sampling, it still costs more training time than the slice or full-text encodings employed in previous studies. Therefore, how to design templates with higher event coding efficiency is another focus of our future work.

## Acknowledgements

The authors would like to thank the three anonymous reviewers for their comments on this paper. This research was supported by the National Natural Science Foundation of China (Nos. 61836007, 62276177, and 62376181), and Project Funded by the Priority Academic Program Development of Jiangsu Higher Education Institutions (PAPD).

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

## A   Undersampling for ECR

To reduce the training cost, we design three undersampling strategies for the ECR task: (1) CorefENN-1 and CorefENN-2: dropping samples that are easy to judge, inspired by ENN (Wilson, 1972), where (i) CorefENN-1: If the top $k$ events with the highest similarity to an event have the same coreference, dropping this event; (ii) CorefENN-2: filtering out the negative samples whose event pair similarity is lower than the threshold $\gamma$; (2) CorefNM: selecting representative negative samples, inspired by NearMiss (Zhang and Mani, 2003). Specifically, for each event, only the top $k$ non-coreferential events with the largest similarity are selected to pair with this event.

Considering that all these strategies exploit the similarity of events, we first build an event encoder based on the Longformer model and attention mechanism, producing similar embeddings for coreferential event mentions. As non-coreferential event mentions may also have similar contexts, we choose Circle Loss (Sun et al., 2020) to optimize the model such that coreferential events would have higher embedding similarities than non-coreferential ones:

$$\mathcal{L}_e = \log \Bigg(1 + \sum_{(ev_i,ev_j)\in\Omega_{pos},(ev_k,ev_l)\in\Omega_{neg}} e^{\lambda\big(\cos(e_k,e_l)-\cos(e_i,e_j)\big)}\Bigg) \tag{11}$$

where $\Omega_{pos}$ and $\Omega_{neg}$ represent coreferential and non-coreferential event sets, respectively, and $e_i, e_j, e_k, e_l$ are event representations.

The statistics of the training data sampled by CorefENN-1, CorefENN-2, and CorefNM are shown in Table 8, where "No" represents the original training set.

| Sampling | Param | Coref | Non-Coref | All |
|---|---|---|---|---|
| No | - | 23,138 | 415,042 | 438,180 |
| CorefENN-1 | k=2 | 23,138 | 16,522 | 39,660 |
| | k=3 | 23,138 | 31,189 | 54,327 |
| | k=4 | 23,138 | 48,726 | 71,864 |
| CorefENN-2 | $\gamma$=0.25 | 23,138 | 13,439 | 36,577 |
| | $\gamma$=0.2 | 23,138 | 23,697 | 46,835 |
| | $\gamma$=0.15 | 23,138 | 42,471 | 65,609 |
| CorefNM | k=2 | 23,138 | 19,300 | 42,438 |
| | k=3 | 23,138 | 28,716 | 51,854 |
| | k=4 | 23,138 | 38,175 | 61,313 |

Table 8: Statistics of the training set constructed by different sampling strategies.

To balance the positive and negative samples and make the datasets obtained by different sampling have similar sizes, we finally set $k$ in CorefENN-1 to 3, $\gamma$ in CorefENN-2 to be 0.2, and set $k$ in CorefNM to be 3. Under these settings, the distribution of events on the long chain, short chain, and singleton is shown in Table 9, where (i) Singleton: at least one event is a singleton; (ii) Long: at least one event comes from a long chain (length > 10), and no event in the event pair is a singleton; (iii) Short: other samples. Here, "Random" represents random undersampling, directly equalizing the number of positive and negative samples.

| Sampling | Singleton | Long | Short |
|---|---|---|---|
| Random | 57.6% | 21.4% | 21.0% |
| CorefENN-1 | 0% | 0.5% | 99.5% |
| CorefENN-2 | 62.5% | 12.2% | 25.3% |
| CorefNM | 71.7% | 5.4% | 22.9% |

Table 9: Event distribution under different sampling.

The filtering of CorefENN-1 drops events from long chains and singletons, resulting in nearly all

| Prompt | Template |
|--------|----------|
| Normal | In the following text, events expressed by `[E1S]` $ev_i$ `[E1E]` and `[E2S]` $ev_j$ `[E2E]` refer to `[MASK]` event:{Segment} |
| Connect | In the following text, the event expressed by `[E1S]` $ev_i$ `[E1E]` `[MASK]` the event expressed by `[E2S]` $ev_j$ `[E2E]`:{Segment} |
| Question | In the following text, do events expressed by `[E1S]` $ev_i$ `[E1E]` and `[E2S]` $ev_j$ `[E2E]` refer to the same event?`[MASK]`.{Segment} |
| Soft | In the following text, `[L1][E1S]` $ev_i$ `[E1E][L2][L3][E2S]` $ev_j$ `[E2E][L4]` `[L5] [MASK][L6]`:{Segment} |

Table 11: Different common templates.

selected events coming from short chains. Both CorefENN-2 and CorefNM focus on sampling non-coreferential event pairs with high similarity; therefore, the proportion of singletons is higher than that of random sampling. The results of using these different undersamplings are shown in Table 10:

| Model | MUC | B3 | CEA. | BLA. | AVG |
|-------|-----|-----|------|------|-----|
| Random | 44.4 | 54.1 | 54.7 | 39.4 | 48.1 |
| CorefENN-1 | 40.0 | 46.8 | 41.8 | 33.1 | 40.4 |
| CorefENN-2 | 44.8 | 56.4 | 58.6 | 40.3 | 50.0 |
| CorefNM | 45.3 | 57.5 | 59.9 | 42.3 | 51.3 |

Table 10: Results with different undersampling.

Table 10 shows that, compared to randomly selecting negative samples, the CorefNM strategy we use selects representative samples based on event similarities, thus achieving a performance improvement of 3.2. CorefENN-1 performs the worst due to missing a large number of events from long chains and singletons, which account for a high proportion of the dataset. Like CorefNM, CorefENN-2 retains non-coreferential event pairs with high similarity; hence, its performance is also higher than random sampling.

## B Computational Efficiency Comparison

Compared with the previous SOTA model **Xu2022** (Xu et al., 2022), our model's advantage is significantly reducing the space complexity without compromising performance. The document-level event encoder used in **Xu2022** (i.e., LongFormer) requires processing the entire document at once, resulting in high GPU memory usage (approximately 44 GB) and necessitating a powerful graphics card (e.g., A100-80G) for operation. This requirement poses an unfriendly challenge to researchers or institutions with limited computing resources. Our segment-level encoder requires only approximately half the memory of the Longformer encoder, about

22 GB, enabling it to run on a standard graphics card (such as the RTX3090).

As our method generates samples for every event mention pair, it requires more training time compared with **Xu2022**. The time cost for **Xu2022** trained on A100-80G is approximately 6.7 hours (4 minutes/epoch, 100 epochs, 400 minutes total), while that of our method on the RTX3090 is about 18 hours (110 minutes/epoch, 10 epochs, 1100 minutes total). However, if we also select the A100-80G for training, the time could be reduced to approximately 9 hours (54 minutes/epoch, 10 epochs, 540 minutes total), similar to that of **Xu2022**.

## C Common Prompts

As shown in Table 11, we construct various common prompts for the ECR task, including (i) Normal: normal template, whose label words directly from the vocabulary, "same" corresponds to coreferential, "different" corresponds to non-coreferential; (ii) Connect: placing the `[MASK]` token between two event mentions, using the descriptions "refer to" and "not refer to" to initialize the embeddings of virtual label words; (iii) Question: asking the PTM whether the two event mentions are coreferential, and the PTM answers "yes" for coreferential and "no" for non-coreferential; (iv) Soft: using learnable tokens that wrap two event mentions to replace the manually written words, as such, the PTM can automatically learn the appropriate template for ECR. The complete form of these templates is shown in Table 11, where Segment represents the original input, and `[L1]` ∼ `[L6]` are learnable tokens.