# OpenReview forum: "CorefPrompt: Prompt-based Event Coreference Resolution by Measuring Event Type and Argument Compatibilities"
_EMNLP/2023/Conference — EMNLP 2023 Main_

### Official Review · Reviewer_TxVr · 2023-07-30

**Soundness:** 2

**Excitement:**

2: Mediocre: This paper makes marginal contributions (vs non-contemporaneous work), so I would rather not see it in the conference.

**Paper Topic And Main Contributions:**

The paper proposes a prompt-based approach for the event coreference resolution (ECR) task, considering compatibilities of event type and argument. The authors conduct an analysis comparing prompt tuning and fine-tuning to show the advantages of prompt-based method on the ECR task.

**Reasons To Accept:**

The analysis and discussion section presents more valuable information.

**Reasons To Reject:**

The motivation of employing the prompt-based method is not clear especially in the introduction section. One might be curious why the prompt-based method is suitable for the ECR task from a technical perspective? It is not necessary to challenge previous studies on points that are proposed in this work unless there is a big improvement in performance causing by those points. But unfortunately, the method only has 0.1 F1 score improvement compared with the SOTA. It is not sufficient to show the effectiveness of this method. Meanwhile, the proposed method costs more training time than the previous methods. In short, I didn't observe many advantages of this work from the modeling perspective. However, the paper looks more like investigating the differences between prompt-based and fine-tuning methods on the ECR task. It will be more interesting to analyse in what cases the methods work well and fail to resolve clusters.

I don't agree the claim presented in the lines 38-40 about entity coreference resolution. Semantic matching is only one of aspects need to be considered in the coreference resolution task. Maybe two mentions have identical semantic meaning, but they could refer to totally different entities. In addition, coreference resolution system also needs to consider compatibilities, such as agreement of number, gender and entity type.

**Reproducibility:**

2: Would be hard pressed to reproduce the results. The contribution depends on data that are simply not available outside the author's institution or consortium; not enough details are provided.

**Reviewer Confidence:**

3: Pretty sure, but there's a chance I missed something. Although I have a good feel for this area in general, I did not carefully check the paper's details, e.g., the math, experimental design, or novelty.

---

> ### Author Rebuttal · Authors · 2023-08-27
>
> Thank you very much for your valuable comments.
>
> **Q1: The motivation of employing the prompt-based method is not clear especially in the introduction section. One might be curious why the prompt-based method is suitable for the ECR task from a technical perspective?**
>
> **A1**: Yes. We will revise the third and the fourth paragraphs to introduce two motivations more clearly as follows:
>
> First, we note that most previous work follows the "encoding-first, then-scoring" framework, wherein they first use encoders (e.g., BERT, SpanBERT, Longformer, etc.) to encode the text and obtain event mention embeddings, and then apply a scorer to evaluate the coreference score of event pairs based on these learned embeddings. This results in the issue of "information blocking." Essentially, since the scorer solely utilizes the learned embeddings as inputs, almost the entire coreference determination relies on the event encoding. However, the event encoding is performed independently, without direct influence from coreference judgment. As a result, the encoder may not accurately capture contextual information that is crucial for identifying event coreference. Especially for coreferential event pairs with unbalanced information, the learned event embeddings may significantly differ if one event has rich arguments (participants) while the other has only a vague event trigger (verb). To address this issue, we convert ECR into a mask language prediction task, using a well-designed prompt. Thus, the event modeling and coreference judgment can conveniently interact based on a fully shared context to improve each other.
>
> Second, in order to restrict the coreference judgment, most previous work has often required designing special model structures to guide the model to focus on the compatibility of event elements. This approach relies on a large amount of training, and cannot guarantee that the model can finally capture the compatibility features or understand the association between compatibilities and coreference. In the worst cases, the automatically captured features focus on other aspects and fail to discover the association between compatibilities and coreference due to the bias of interaction features or noise in data. In this paper, we introduce two auxiliary prompt tasks, event-type compatibility and argument compatibility, to explicitly demonstrate the inference process of coreference judgment in the template and guide the model to make final decisions based on these compatibilities. Taking advantage of the convenience of incorporating human knowledge into a prompt-based approach, our method can navigate the model's focus on event type and argument compatibilities using templates in natural language. This not only utilizes the discriminant rules outlined by annotators, but also adjusts to different levels of compatibility with the assistance of soft label words.
>
> **Q2: The method only has 0.1 F1 score improvement compared with the SOTA. It is not sufficient to show the effectiveness of this method. Meanwhile, the proposed method costs more training time than the previous methods.**
>
> **A2**: Compared with the previous SOTA model by Xu et al. (2022), our model's advantage is significantly reducing space complexity without compromising performance. The document-level event encoder used in SOTA (i.e., LongFormer) requires processing the entire document at once, resulting in high GPU memory usage (approximately 44GB) and necessitating a powerful graphics card (e.g., A100-80G) for operation. This requirement poses an unfriendly challenge to researchers or institutions with limited computing resources. Our segment-level encoder requires only approximately half the memory of Longformer, approximately 22GB, enabling it to run on a standard graphics card (such as the RTX 3090).
>
> As our method generates samples for every event mention pair, it requires more training time. The time cost for the previous SOTA model (Xu et al. 2022) trained on A100-80G is approximately 6.7 hours (4 minutes/epoch, 100 epochs, 400 minutes total), while that of our method on the RTX3090 is approximately 18 hours (110 minutes/epoch, 10 epochs, 1100 minutes total). However, if we select the A100-80G for training, the time could be reduced to approximately 9 hours (54 minutes/epoch, 10 epochs, 540 minutes total), similar to that of the SOTA.
>
> We will include these comparisons in the revised version.
>
> Moreover, compared to the previous SOTA (Lu and Ng, 2021) that also employs segment-level context, our approach shows a significant improvement, with an average F1 score increase of 3.3. Our approach can be seen as a replacement for document-level event encoding to some extent, as we compensate for the limitations of segment-level event encoding by integrating human guidance and leveraging the knowledge embedded in PTM. This permits our model to retain exceptional performance while notably decreasing GPU memory usage.
>
> **Q3: I don't agree the claim presented in the lines 38-40 about entity coreference resolution. Semantic matching is only one of aspects need to be considered in the coreference resolution task. In addition, coreference resolution system also needs to consider compatibilities, such as agreement of number, gender and entity type.**
>
> **A3**: Yes. Ambiguity arises here due to inaccuracies in our descriptions. In fact, we want to express that event coreference resolution is more challenging than entity coreference resolution due to the complex event structures, i.e., triggers and corresponding arguments are loosely distributed in the text. We will fix this problem in the revised paper.

---

### Official Review · Reviewer_PPJ5 · 2023-08-05

**Soundness:** 4

**Excitement:**

4: Strong: This paper deepens the understanding of some phenomenon or lowers the barriers to an existing research direction.

**Paper Topic And Main Contributions:**

This paper is about a novel solution to the event coreference resolution problem. The paper proposes a prompt-based approach Coref-Prompt to transform ECR into a cloze-style MLM (Masked Language Model) task.

**Questions For The Authors:**

- Granted that you produce a novel methodology using prompt tuning, but how do you know this Large Language Model has not seen the testing data during its training over the entire internet-crawled data? I mean, this is a pertinent deeper issue with the current state of the art, but my point is you must at east mention that as a limitation.



**Reasons To Accept:**

The paper introduces a novel methodology for Event Coreference Resolution and proves empirically that the solution is better than SOTA. The solution idea is novel, uses state of the art, and is a thoroughly written comprehensive paper from an engineering perspective considering all possible evaluation methodologies.

**Reasons To Reject:**

- Core solution idea is probably good however the paper is not at all well written/motivated/it is not a smooth reading experience. The paper should not be accepted in this form/ needs a thorough revamping in the first 2 pages, despite this being a thorough engineering work

Abstract
- Line 022- what is KBP 2017. You are assuming every reader knows what it is. Further, in abstract you don't need to mention specific datasets. Just say performs well in a state of the art benchmark.

Introduction

- Introduction section does not read like an introduction and more like a related work section. Does not motivate the problem well

- Can you add one solid example of event coreference resolution in Introduction. Maybe at line 27 example. 1) Not everyone who is going to read this paper is going to know what it is 2) a research paper needs to be self contained as much as possible.

E.g. Like the one in the introduction section of this work: Lu, Jing, and Vincent Ng. "Span-based event coreference resolution." Proceedings of the AAAI conference on artificial intelligence. Vol. 35. No. 15. 2021.

- You keep mentioning encoding first and then scoring, but does not explain anywhere what it means. Refer comment above about paper being self contained

- Line 121- again, your method performing well is not your contribution. That is the result.

- Line 413- you throw a ton of names without any reference or explanation

- Line 475- i don't understand why you say comparable performance. Your work actually beats the SOTA with a good margin.

Model:

You suddenly jump into a bunch of math/minor details without explaining well what the model does in plain text/no jargon. Refer to the motivation issue mentioned above which seems to be a pertinent theme throughout the paper. This currently reads as a pure engineering paper and not an NLP paper.



**Reproducibility:**

4: Could mostly reproduce the results, but there may be some variation because of sample variance or minor variations in their interpretation of the protocol or method.

**Reviewer Confidence:**

3: Pretty sure, but there's a chance I missed something. Although I have a good feel for this area in general, I did not carefully check the paper's details, e.g., the math, experimental design, or novelty.

**Typos Grammar Style And Presentation Improvements:**


-Figure 1: is way too much information without absolutely any explanation whatsoever. A figure and its caption together must be completely self contained.

- Line 111- what is KBP 2017. Either add a citation. Or don't mention the results in Intro. You don't need to talk about the effectiveness of your work in intro- where you just have to introduce the problem thoroughly, which you don't.

---

> ### Author Rebuttal · Authors · 2023-08-27
>
> Thank you very much for your valuable comments.
>
> **Q1: In abstract you don't need to mention specific datasets. Just say performs well in a state of the art benchmark. Line 121 your method performing well is not your contribution. That is the result.**
>
> **A1**: Yes, we will revise the last sentence of the abstract as follows:
>
> Experimental results show that our method CorefPrompt performs well in a state-of-the-art benchmark.
>
> We will also delete “Our method is simple yet effective, achieving comparable performance to SOTA.” from our contributions.
>
> **Q2: Introduction section does not read like an introduction and more like a related work section. Does not motivate the problem well. You keep mentioning encoding first and then scoring, but does not explain anywhere what it means.**
>
> **A2**: Yes. We will revise the third and the fourth paragraphs to introduce two motivations more clearly as follows:
>
> First, we note that most previous work follows the "encoding-first, then-scoring" framework, wherein they first use encoders (e.g., BERT, SpanBERT, Longformer, etc.) to encode the text and obtain event mention embeddings, and then apply a scorer to evaluate the coreference score of event pairs based on these learned embeddings. This results in the issue of "information blocking." Essentially, since the scorer solely utilizes the learned embeddings as inputs, almost the entire coreference determination relies on the event encoding. However, the event encoding is performed independently, without direct influence from coreference judgment. As a result, the encoder may not accurately capture contextual information that is crucial for identifying event coreference. Especially for coreferential event pairs with unbalanced information, the learned event embeddings may significantly differ if one event has rich arguments (participants) while the other has only a vague event trigger (verb). To address this issue, we convert ECR into a mask language prediction task, using a well-designed prompt. Thus, the event modeling and coreference judgment can conveniently interact based on a fully shared context to improve each other.
>
> Second, in order to restrict the coreference judgment, most previous work has often required designing special model structures to guide the model to focus on the compatibility of event elements. This approach relies on a large amount of training, and cannot guarantee that the model can finally capture the compatibility features or understand the association between compatibilities and coreference. In the worst cases, the automatically captured features focus on other aspects and fail to discover the association between compatibilities and coreference due to the bias of interaction features or noise in data. In this paper, we introduce two auxiliary prompt tasks, event-type compatibility and argument compatibility, to explicitly demonstrate the inference process of coreference judgment in the template and guide the model to make final decisions based on these compatibilities. Taking advantage of the convenience of incorporating human knowledge into a prompt-based approach, our method can navigate the model's focus on event type and argument compatibilities using templates in natural language. This not only utilizes the discriminant rules outlined by annotators, but also adjusts to different levels of compatibility with the assistance of soft label words.
>
> We will also reorganize the introduction part of previous work around these two issues.
>
> **Q3: Can you add one solid example of event coreference resolution in Introduction. Figure 1: is way too much information without absolutely any explanation whatsoever.**
>
> **A3**: Yes. We will add an example in the Introduction section to explain what event coreference resolution is as follows, where "[]" is used to tag the annotated spans and "{}" is used to tag annotated labels.
>
> [Former Pakistani dancing girl]{arg1} commits [suicide]{ev1} 12 years after horrific [acid]{arg2} [attack]{ev2} which [left]{ev3} [her]{arg3} looking 'not human'. [She]{arg4} had undergone 39 separate surgeries to repair [damage]{ev4}. Leapt to [her]{arg5} [death]{ev5} from [sixth floor Rome building]{arg6} [earlier this month]{arg7}. [Her ex-husband]{arg8} was [charged]{ev6} with [attempted murder]{arg9} in [2002]{arg10} but has since been [acquitted]{ev7}.
>
> The example above contains seven event mentions (marked as ev1-7) and ten entities (marked arg1-11) that serve as arguments. Among them, the die event mention ev1 with the argument arg1 and the die event mention ev5 with the arguments arg5, arg6, and arg7 are coreferential, and both of them describe the girl's suicide by jumping off a building; the injury event mention ev3 with the arguments arg2 and arg3 and the injury event ev4 with the argument arg4 are coreferential, and both of them describe the girl's disfigurement; other events are singletons.
>
> We will also show the specific template of this example in the Model section corresponding to Figure 1.
>
> **Q4: Line 413 you throw a ton of names without any reference or explanation.**
>
> **A4**: We inadvertently omitted important citations on the KBP corpus in Section 4.1. In the revised version, we will add these citations [Mitamura et al., 2015, 2016, 2017] and give a more detailed introduction to the KBP dataset.
>
> [Mitamura et al., 2015] Teruko Mitamura, Zhengzhong Liu, and Eduard H Hovy. 2015. Overview of TAC KBP 2015 event nugget track. In Proceedings of the Text Analysis Conference.
> [Mitamura et al., 2016] Teruko Mitamura, Zhengzhong Liu, and Eduard H Hovy. 2016. Overview of TAC KBP 2016 event nugget track. In Proceedings of the Text Analysis Conference.
> [Mitamura et al., 2017] Teruko Mitamura, Zhengzhong Liu, and Eduard H Hovy. 2017. Overview of TAC KBP 2015 event nugget track. In Events Detection, Coreference and Sequencing: What’s Next? Overview of the TAC KBP 2017 Event Track.
>
> **Q5: You suddenly jump into a bunch of math or minor details without explaining well what the model does in plain text/no jargon. Refer to the motivation issue mentioned above which seems to be a pertinent theme throughout the paper. Figure 1: is way too much information without absolutely any explanation whatsoever.**
>
> **A5**: Yes. Our introduction to the model needs better organization. Specifically, in lines 192-206, we merged the introduction to prompting with our method, which is only a component of our entire framework. In the revised version, we will create a subsection exclusively focusing on introducing prompting. Secondly, we have not comprehensively described our method regarding motivations. To address this, we will include the following paragraph, demonstrating how our model in Figure 1 can tackle the two motivations mentioned in Section 1.
>
> For the input event mention pair (e_i, e_j), we construct three corresponding templates: the prefix template T_pre, the anchor template T_anc, and the inference template T_inf. These templates respectively add guidance, encode events (including predicting event types), and discriminate coreference (including judging type and argument compatibilities). Then, we embed these templates into the segment hosting the two event mentions, converting the sample into a prompt containing multiple [mask] tokens. Since all event tasks are completed simultaneously in the same template, multiple steps, such as event modeling and coreference judgment, can interact conveniently based on a fully shared context, reducing the "information blocking" problem that existed in previous work. Finally, we send the entire prompt to the PTM and use the PTM's MLM head to predict the results of all tasks. We guide the PTM's encoding with the prefix template and explicitly demonstrate the coreference judgment reasoning process with the inference template, fully incorporating human knowledge into the model's predictions.
>
> The caption for Figure 1 is too brief, so we will add a simplified version of the complete process to the caption.
>
> **Q6: How do you know this Large Language Model has not seen the testing data during its training over the entire internet-crawled data? My point is you must at east mention that as a limitation.**
>
> **A6**: Yes, in theory, all pre-trained models carry the risk of information leakage, which we will address in the limitations section. However, our model solely employs RoBERTa, one of the traditional PTMs. Therefore, this issue is not significant, given that traditional PTMs' (such as BERT and RoBERTa) pre-training tasks don't involve entity or event coreference resolution. While the pre-training process for popular LLMs such as GPT3 and LLaMa may involve entity coreference or event extraction, their risk of information leakage will significantly increase.
>
> **Q7: Line 475- I don't understand why you say comparable performance. Your work actually beats the SOTA with a good margin.**
>
> **A7**: You are right. Our model CorefPrompt outperforms the SOTA by 0.1 on AVG.

---

### Official Review · Reviewer_B9w6 · 2023-08-10

**Soundness:** 4

**Excitement:**

4: Strong: This paper deepens the understanding of some phenomenon or lowers the barriers to an existing research direction.

**Paper Topic And Main Contributions:**

This paper presents a novel approach to tackling within-document Event Coreference Resolution (ECR). The authors achieve this by reimagining the coreference prediction task as a cloze-style Masked Language Model (MLM) task. Through this technique, they effectively merge ECR with event type extraction and the prediction of event type and argument compatibility within the MLM framework using carefully designed prompt templates. This integration seemingly facilitates their system to adeptly grasp the intricate rationale behind event types and arguments compatibilities when making coreference decisions. They finally achieve near SOTA results for a specific dataset with this methodology while only using the segments instead of the entire document.

**Questions For The Authors:**

Question A: What are the compute requirements? Training and inference times?

**Reasons To Accept:**

- nice novel design bringing together ideas from various different previous works, and a solid experimental setup
- comprehensive ablation study
- well-written and clear (for an audience familiar with this work)
- serves as a basis for modeling other event features as cloze task

**Reasons To Reject:**

- There were no proper justifications given about the method's simplicity when compared to previous SOTA method. Is it more computationally efficient than previous work? Why use only segment when the previous method got a big bump in performance with the entire document? Without the justifications, using only segments in this work seems wasteful.
- While the joint modeling of event types with coreference is ingenious, there is a limitation the paper puts itself in when tackling new event types. It seems like their methodology would be more far-reaching (on more than 1 dataset) if the authors could address how they would tackle new event types in this paper. Since they don't do it, it seems like a missed opportunity.

**Reproducibility:**

4: Could mostly reproduce the results, but there may be some variation because of sample variance or minor variations in their interpretation of the protocol or method.

**Reviewer Confidence:**

4: Quite sure. I tried to check the important points carefully. It's unlikely, though conceivable, that I missed something that should affect my ratings.

**Typos Grammar Style And Presentation Improvements:**

-  line 06: relies -> rely
- line 026: Why not just call it Within-document Event Coreference Resolution? changing terminology might throw off the readers
- Only English corpora experiments should be a limitation
- The paper in general should be made clearer for audiences who are not familiar with this work. Suggestions include adding examples (textual) in Figure 1, additional details about the corpora used, an explanation of how training and inference algorithms are different, etc.

---

> ### Author Rebuttal · Authors · 2023-08-27
>
> Thank you very much for your valuable comments.
>
> **Q1: There were no justifications given about the method's simplicity when compared to previous SOTA method. Why use only segment? What are the compute requirements? Training and inference times?**
>
> **A1**: Compared with the previous SOTA model by Xu et al. (2022), our model's advantage is significantly reducing space complexity without compromising performance. The document-level event encoder used in SOTA (i.e., LongFormer) requires processing the entire document at once, resulting in high GPU memory usage (approximately 44GB) and necessitating a powerful graphics card (e.g., A100-80G) for operation. This requirement poses an unfriendly challenge to researchers or institutions with limited computing resources. Our segment-level encoder requires only approximately half the memory of Longformer, approximately 22GB, enabling it to run on a standard graphics card (such as the RTX 3090).
>
> As our method generates samples for every event mention pair, it requires more training time. The time cost for the previous SOTA model (Xu et al. 2022) trained on A100-80G is approximately 6.7 hours (4 minutes/epoch, 100 epochs, 400 minutes total), while that of our method on the RTX3090 is approximately 18 hours (110 minutes/epoch, 10 epochs, 1100 minutes total). However, if we select the A100-80G for training, the time could be reduced to approximately 9 hours (54 minutes/epoch, 10 epochs, 540 minutes total), similar to that of the SOTA.
>
> We will include these comparisons in the revised version.
>
> **Q2: There is a limitation the paper puts itself in when tackling new event types. It seems like their methodology would be more far-reaching if the authors could address how they would tackle new event types in this paper.**
>
> **A2**: Yes. Event coreference resolution with open event types is more aligned with real-world scenarios, which is a focus of our future work. However, since event coreference resolution is challenging (even though the performance of recent state-of-the-art models is still far from practical), most current research is carried out in scenarios with fixed event types. Therefore, to compare with the state-of-the-art models, we adhere to the setting of fixed event types.
>
> In addition, our model can handle unseen event types to some extent by converting multiple event tasks into MLM tasks. During prediction, all task positions except for coreference judgment, such as event type prediction, need only be kept as [MASK] (Line 403-408) in the model. Therefore, while our prompting method has only been trained on the KBP dataset with fixed event types, the pre-trained model (PTM) may autonomously predict these mask tokens as corresponding label words (other words in the vocabulary) when encountering unseen event types, drawing on its own knowledge.
>
> **Q3: Only English corpora experiments should be a limitation.**
>
> **A3**: Yes, evaluating our model across various languages can establish its generality. However, while the present prominent KBP datasets comprise non-English languages such as Chinese and Spanish, recent studies in this field have concentrated solely on English due to the small size of non-English data. For instance, the number of Chinese documents in both the KBP 2016 and 2017 datasets is merely 334, which only accounts for approximately one-third of the English data size, and is inadequate to support contemporary deep learning techniques.
>
> **Q4: The paper should be made clearer for audiences who are not familiar with this work. Suggestions include adding examples, additional details about the corpora, etc.**
>
> **A4:** Yes. We will add an example in the Introduction section to explain what event coreference resolution is as follows, where "[]" is used to tag the annotated spans and "{}" is used to tag annotated labels.
>
> [Former Pakistani dancing girl]{arg1} commits [suicide]{ev1} 12 years after horrific [acid]{arg2} [attack]{ev2} which [left]{ev3} [her]{arg3} looking 'not human'. [She]{arg4} had undergone 39 separate surgeries to repair [damage]{ev4}. Leapt to [her]{arg5} [death]{ev5} from [sixth floor Rome building]{arg6} [earlier this month]{arg7}. [Her ex-husband]{arg8} was [charged]{ev6} with [attempted murder]{arg9} in [2002]{arg10} but has since been [acquitted]{ev7}.
>
> The example above contains seven event mentions (marked as ev1-7) and ten entities (marked arg1-11) that serve as arguments. Among them, the die event mention ev1 with the argument arg1 and the die event mention ev5 with the arguments arg5, arg6, and arg7 are coreferential, and both of them describe the girl's suicide by jumping off a building; the injury event mention ev3 with the arguments arg2 and arg3 and the injury event ev4 with the argument arg4 are coreferential, and both of them describe the girl's disfigurement; other events are singletons.
>
> We will also show the specific template of this example in the Model section corresponding to Figure 1.
>
> We inadvertently omitted important citations on the KBP corpus in Section 4.1. In the revised version, we will add these citations [Mitamura et al., 2015, 2016, 2017] and give a more detailed introduction to the KBP dataset.
>
> [Mitamura et al., 2015] Teruko Mitamura, Zhengzhong Liu, and Eduard H Hovy. 2015. Overview of TAC KBP 2015 event nugget track. In Proceedings of the Text Analysis Conference.
> [Mitamura et al., 2016] Teruko Mitamura, Zhengzhong Liu, and Eduard H Hovy. 2016. Overview of TAC KBP 2016 event nugget track. In Proceedings of the Text Analysis Conference.
> [Mitamura et al., 2017] Teruko Mitamura, Zhengzhong Liu, and Eduard H Hovy. 2017. Overview of TAC KBP 2015 event nugget track. In Events Detection, Coreference and Sequencing: What’s Next? Overview of the TAC KBP 2017 Event Track.

---

### Meta-Review · Area_Chair_c8WP · 2023-09-19

**Recommendation:** 4

**Metareview:**

The reviews have not reached a consensus on both soundness and excitement.

Regarding soundness, there is a discrepancy in whether the empirical observations (especially Table 1) adequately support the effectiveness of the proposed method.  I see the effectiveness is well verified, because the proposed method uses segment-level contexts and outperforms state-of-the-art  (Lu and Ng, 2021b) using segment-level contexts, while rivaling state-of-the-art (Xu et al., 2022) using document-level contexts.

As for excitement, the proposed prompt-based method for within-document event coreference resolution is novel.  I would value this novelty, because to my knowledge this paper is the first paper that employs prompting techniques for event coference resolution.

---

### Decision · Program_Chairs · 2023-10-07

**Decision:**

Accept-Main

**Comment:**

The reviews have not reached a consensus on both soundness and excitement.

Regarding soundness, there is a discrepancy in whether the empirical observations (especially Table 1) adequately support the effectiveness of the proposed method.  I see the effectiveness is well verified, because the proposed method uses segment-level contexts and outperforms state-of-the-art  (Lu and Ng, 2021b) using segment-level contexts, while rivaling state-of-the-art (Xu et al., 2022) using document-level contexts.

As for excitement, the proposed prompt-based method for within-document event coreference resolution is novel.  I would value this novelty, because to my knowledge this paper is the first paper that employs prompting techniques for event coference resolution.